# Real-Time Positron Emission Tomography Evaluation of Topotecan Brain Kinetics after Ultrasound-Mediated Blood–Brain Barrier Permeability

**DOI:** 10.3390/pharmaceutics13030405

**Published:** 2021-03-18

**Authors:** Andrei Molotkov, Patrick Carberry, Martin A. Dolan, Simon Joseph, Sidney Idumonyi, Shunichi Oya, John Castrillon, Elisa E. Konofagou, Mikhail Doubrovin, Glenn J. Lesser, Francesca Zanderigo, Akiva Mintz

**Affiliations:** 1Department of Radiology, Columbia University Medical Center, 722 West 168th Street, New York, NY 10032, USA; am3355@cumc.columbia.edu (A.M.); pc2545@cumc.columbia.edu (P.C.); md3874@cumc.columbia.edu (M.A.D.); sj2244@columbia.edu (S.J.); si2302@cumc.columbia.edu (S.I.); so267@cumc.columbia.edu (S.O.); jc944@cumc.columbia.edu (J.C.); md2367@cumc.columbia.edu (M.D.); 2Department of Biomedical Engineering, Columbia University Medical Center, 722 West 168th Street, New York, NY 10032, USA; ek2191@columbia.edu; 3Department of Internal Medicine, Section on Hematology and Oncology, Wake Forest Baptist Comprehensive Cancer Center, Winston-Salem, NC 27157, USA; glesser@wakehealth.edu; 4Department of Psychiatry, Columbia University Medical Center, 722 West 168th Street, New York, NY 10032, USA; Francesca.Zanderigo@nyspi.columbia.edu; 5Molecular Imaging and Neuropathology Area, New York State Psychiatric Institute, New York, NY 10032, USA

**Keywords:** blood–brain barrier, high-intensity focused ultrasound, positron emission tomography, topotecan, glioblastoma

## Abstract

Glioblastoma (GBM) is the most common primary adult brain malignancy with an extremely poor prognosis and a median survival of fewer than two years. A key reason for this high mortality is that the blood–brain barrier (BBB) significantly restricts systemically delivered therapeutics to brain tumors. High-intensity focused ultrasound (HIFU) with microbubbles is a methodology being used in clinical trials to noninvasively permeabilize the BBB for systemic therapeutic delivery to GBM. Topotecan is a topoisomerase inhibitor used as a chemotherapeutic agent to treat ovarian and small cell lung cancer. Studies have suggested that topotecan can cross the BBB and can be used to treat brain metastases. However, pharmacokinetic data demonstrated that topotecan peak concentration in the brain extracellular fluid after systemic injection was ten times lower than in the blood, suggesting less than optimal BBB penetration by topotecan. We hypothesize that HIFU with microbubbles treatment can open the BBB and significantly increase topotecan concentration in the brain. We radiolabeled topotecan with ^11^C and acquired static and dynamic positron emission tomography (PET) scans to quantify [^11^C] topotecan uptake in the brains of normal mice and mice after HIFU treatment. We found that HIFU treatments significantly increased [^11^C] topotecan brain uptake. Moreover, kinetic analysis of the [^11^C] topotecan dynamic PET data demonstrated a substantial increase in [^11^C] topotecan volume of distribution in the brain. Furthermore, we found a decrease in [^11^C] topotecan brain clearance, confirming the potential of HIFU to aid in the delivery of topotecan through the BBB. This opens the potential clinical application of [^11^C] topotecan as a tool to predict topotecan loco-regional brain concentration in patients with GBMs undergoing experimental HIFU treatments.

## 1. Introduction

Glioblastoma (GBM) is the most common primary adult brain tumor with an extremely poor prognosis and median survival of less than 2 years [1,2]. One key reason for this high mortality is that the blood–brain barrier (BBB) significantly restricts the targeted delivery of therapeutics to brain tumors [3]. Although the core of the GBM has a leaky BBB that is permeable to magnetic resonance imaging (MRI) contrast agents, allowing it to be successfully imaged, areas of tumor cell infiltration that exist outside this contrast enhancing center are not subject to effective concentrations of systemically administered therapies. High-intensity focused ultrasound (HIFU) is a noninvasive method of permeabilizing the BBB using ultrasound waves in conjunction with systemically administered microbubbles and has shown promise in preclinical and clinical trials [4,5,6]. Several factors have been shown to play a role in increasing BBB permeability after HIFU. The primary mechanism is thought to be secondary to microbubble oscillations in the blood vessels, which create shear stress and the rapid collapse of microbubbles, resulting in decreased tight junction integrity [7]. When applied to in vitro models of the BBB, HIFU has also been shown to have the ability to pull monolayers of the cell membrane apart and create air pockets to enhance drug permeability [8]. In addition to the physical effects of HIFU, it has been shown that the combination of ultrasound and microbubbles resulted in a damage-associated molecular pattern (DAMP) response, which elevated a number of inflammatory proteins, including heat-shock protein 70, IL-1, IL-18, and TNFα [9]. This inflammatory response has been shown to be driven by the induction of the NFĸB pathway. We and others have shown that the presence of inflammatory cytokines can independently permeabilize the BBB [10,11]. Thus, HIFU, in conjunction with microbubbles, is an emerging method that can potentially enable drugs to reach the infiltrating GBM cells outside the naturally BBB-permeable core [4,10,12,13,14,15].

Topotecan is a topoisomerase inhibitor used as a chemotherapeutic agent to treat ovarian and small cell lung cancer (SCLC) [16]. Some studies suggest that topotecan crosses the BBB [17] and can be used to treat SCLC brain metastases [18,19]. However, pharmacokinetic data in rats demonstrated that the topotecan peak concentration in the brain extracellular fluid was ten times lower than in the blood after intravenous (i.v.) injection [17]. Furthermore, clinical studies demonstrated that despite topotecan reaching cytotoxic concentrations (>1 ng/mL) in the brain after systemic injection, the systemically delivered dose required caused severe hematological toxicity [18]. Despite its limitations when administered intravenously, topotecan is actively being explored as a therapeutic against GBM when delivered with convection-enhanced delivery (CED). Preclinical data in mouse and rat orthotopic gliomas demonstrated tumor regression and increased survival in rodents infused with topotecan [20,21]. These data demonstrate the potential of topotecan to effectively eradicate both tumor cells and glial progenitor cells. In a recent study, Upadhyayula et al., reported the initial experience using topotecan in 10 GBM and 6 anaplastic astrocytoma patients delivered via CED. The treatment was deemed tolerable and demonstrated initial signs of efficacy, with a reported 20% two-year survival rate for GBM patients, including two patients that survived over ten years, a rarity for GBM patients [22].

Our overall goal of this study is to use positron emission tomography (PET) imaging of [^11^C] topotecan after treatment with HIFU and microbubbles to demonstrate that we can permeabilize the BBB and increase effective topotecan loco-regional concentrations in the brain. We hypothesize that the combination of HIFU with microbubbles and topotecan administration is a promising treatment strategy for the therapy of brain malignancies.

## 2. Material and Methods

### 2.1. General Information

Cassettes, reagent kits (Prod No. PE-FSPG-047-R), the precursor (Prod No. 3193.0075), elution solution (Prod No. PE-FSPG-047-R-V1), and reference standard (PE-FSPG-047-H) were purchased from ABX (Radeberg, Germany). *N*-Desmethyl topotecan was purchased from Synfine Research (Toronto, Canada). Topotecan hydrochloride hydrate (>98%) standard and *N*,*N*-dimethylformamide (DMF, anhydrous 99.8%) were purchased from Sigma-Aldrich (St. Louis, MO, USA) and used without further purification. Sodium hydroxide National Institute of Standards and Technology (NIST) traceable solution (0.5 N) was purchased from Aqua Solutions, Inc. (Deer Park, TX, USA) and used without further purification. Solid-phase extraction cartridges were purchased from Waters (Milford, MA, USA). Ultra-high purity N.O.S. gas (99% nitrogen/1% oxygen) was purchased from Airgas (White Plains, NY, USA). All other reagents not listed above were of the highest grade available from Sigma-Aldrich (St. Louis, MO, USA) and Fisher Scientific (Pittsburgh, PA, USA). End of synthesis radioactivity was determined using a Biodex AtomLab 500 dose calibrator (Shirley, NY, USA).

### 2.2. Radiosynthesis of [^18^F]-(4S)-4-(3-[18F]-fluoropropyl)-L Glutamic Acid ([^18^F] FSPG)

^18^F was produced on the Siemens Eclipse HP cyclotron RDS 111 (11-MeV) in fluorine radioisotope nuclear reaction ^18^O(p,n)^18^F. The synthesis was performed as previously reported [23] on the ORA Neptis Synthesizer Unit. Briefly, 6 mg of FSPG precursor, di-*tert*-butyl (4S)-4-(3-((2-naphthylsulfonyl)oxy)propyl)-*N*-trityl-l-glutamate, was transferred to a 10 mL vial with 1.4 mL of acetonitrile, mixed, and loaded into the reaction vessel. Siemens Eclipse cyclotron was loaded with 2.4 mL of [^18^O] enriched water and was bombarded for 30 min at 55 μA. To produce [^18^F] FSPG, ^18^F solution was loaded onto a QMA light cartridge (Neptis platform) and eluted with 800 μL mixture of 0.5 M potassium carbonate in water and 4,7,13,16,21,24-hexaoxa-1,10-diazabicyclo [8.8.8] hexacosane (cryptand 222) in acetonitrile. Following drying of the cryptand 222 [^18^F] complex, the FSPG precursor was labeled with [^18^F]. The reaction was heated to 130 °C for 8 min, and the crude radiolabeled intermediate was de-protected with the 6 mL of 1M sulfuric acid at 70 °C, followed by neutralization with 1.5 mL of 4M sodium hydroxide. [^18^F] FSPG was purified by loading onto a solid-phase extraction cartridge (two MCX cartridges) and washing with 10 mL of water. After washing, [^18^F] FSPG was eluted with 20 mL of buffer solution (disodium hydrogen phosphate dehydrate, saline, and sterile water for injection) through an Alumina N cartridge attached to a 500 mg Hypercarb cartridge. The automated synthesis time was about 45 min to provide 9250 MBq (250 mCi) of [^18^F] FSPG (decay corrected yield = 33%).

### 2.3. Radiosynthesis of [^11^C] Topotecan ([^11^C]-(S)-10-[(Dimethylamino)methyl]-4-ethyl-4,9-dihydroxy-1H pyrano[3’,4′,:6,7]indolinizo[1,2,b] Quinolone-3,14 (4H,12H)-Dionemonohydrochloride)

Carbon-11 labeled topotecan was synthesized using a modification of the previously reported scheme by Yamasaki et al. [24]. Briefly, desmethyl topotecan (0.4 mg) was dissolved in anhydrous DMF (300 μL) and 0.5 N sodium hydroxide (5 μL). A Siemens Eclipse HP cyclotron RDS 111 (11-MeV) was bombarded for 30 min at 50 μA, and the radioactive carbon-11 carbon dioxide [^11^C] CO_2_ gas was converted to [^11^C] iodomethane via GE F_x_MeI module. The [^11^C] iodomethane gas was bubbled into the reaction mixture at room temperature. Once activity had plateaued, the reaction vessel was moved to a heating block set at 80 °C and heated for 5 min. The reaction was then removed from heat and quenched with 300 μL of semi-prep mobile phase and taken up into a high-performance liquid chromatography (HPLC) loop. The crude product was purified by way of HPLC using a mobile phase consisting of methanol/water/trifluoroacetic acid (40/60/0.06) at a flow rate of 4.0 mL/min. The desired fraction was collected (r_t_~8 min) and loaded onto a pre-conditioned *t*-C_18_ plus cartridge (conditioned via 10 mL ethanol followed by 10 mL of sterile water for injection) with the use of 100 mL of sterile water for injection. The cartridge was washed with 10 mL of sterile water for injection and then eluted to a final product vial through a sterile GVM 0.22-μm filter with the use of 1.0 mL ethanol followed by 9 mL of 0.9% saline to afford [^11^C] topotecan (42-min post start of synthesis (SOS)), a pale-yellow solution. The radiochemical purity of [^11^C] topotecan determined by HPLC analysis was 94 ± 3% (*n* = 4).

### 2.4. Cell Culture

Human glioblastoma (GBM) G48a and U251 cells (American Type Culture Collection (ATCC), Manassas, VA, USA) were cultured in DMEM basal media supplemented with 10% of heat-inactivated fetal bovine serum (FBS) (Gibco, Dublin, Ireland), 2 mM glutamine, and 1/200 of penicillin-streptomycin mix. Topotecan (T2705, Sigma, St. Louis, MO, USA) was added to the cell culture medium at 0.01–1 μg/mL for 24 h. Cell proliferation was measured using CyQUANT assay (C7026, ThermoFisher, Rochester, NY, USA) on a Spark plate reader (Tecan, Switzerland).

### 2.5. Mice

C57BL/6NTac mice (B6, Taconic) were maintained on a normal mouse diet. All animal experiments were conducted according to protocols approved by the Institutional Animal Care and Use Committee of Columbia University Medical Center (AC-AAAT8456 (25 January 2021)).

### 2.6. HIFU with Microbubbles

DEFINITY^®^ microbubbles were prepared according to manufacturer instructions (Lantheus Medical Imaging, Inc., Billerica, MA, USA). HIFU was applied through the intact skin to the right cerebral hemisphere using the RK50 system (FUS Instruments, Toronto, ON, Canada). Ultrasound was delivered as a series of burst exposures (10 ms duration, 1 s repetition frequency, ultrasound frequency of 1.5 MHz, and 0.7 MPa peak negative focal pressure). Simultaneous with the start of HIFU application, DEFINITY^®^ microbubbles (4.5 × 10^7^ microbubbles in 0.5 mL of 0.9% NaCl sterile solution) were infused through a catheter at a flow rate of 50 μL/min into a mouse-tail vein [25]. Immediately after HIFU treatment, 75 μL of Cy7-albumin (Cy7 ALB) (2.6 mg/mL) and 50 μL of 2% Evan’s blue (EB) solution were injected through the same tail vein catheter to verify the opening of the BBB.

### 2.7. PET Experiments

[^18^F] FSPG: 10–20 min after HIFU treatment, B6 mice were injected i.v. with 4.7–5.7 MBq (128–153 μCi) of [^18^F] FSPG. 30 min post [^18^F] FSPG injection, mice were placed into a 4-mouse bed, and 30 min static PET images were acquired using a micro PET scanner (Siemens, Munich, Germany) followed by microCT (MIlabs, Houten, The Netherlands) on the same bed for anatomical reference.

[^11^C] topotecan: 10–20 min post-HIFU treatment, B6 mice were placed into a 4-mouse bed and injected i.v. with 5.4–7.9 MBq (146–214 μCi) of [^11^C] topotecan. 60 min dynamic PET images were acquired using a micro PET scanner (Siemens, Munich, Germany) followed by microCT (MIlabs, Houten, The Netherlands) on the same bed for anatomical reference.

For both [^18^F] FSPG and [^11^C] topotecan studies, immediately after PET/CT scans, mice were euthanized, and their dissected brains were placed into 4% PFA and imaged for 10 min with static PET. Regions of interest (ROI) were manually drawn over the heart and the right brain hemisphere around the area targeted by HIFU using microCT as a reference. All PET images were reconstructed using the 3D-OSEM algorithm with 3-iterations in 256 × 256 matrix (Inveon, Siemens, Munich, Germany) and analyzed using VivoQuant ver 4 (Invicro, Boston, MA, USA).

### 2.8. [^11^C] Topotecan Kinetic Analysis

The [^11^C] topotecan radioactivity curve over time in the heart ROI was used as a proxy for the arterial input function during the kinetic modeling of the tracer time–activity curves (TACs) in the brain ROI. In both groups of control mice (mice not treated with HIFU, *n* = 4) and mice after HIFU with microbubbles treatment (*n* = 4), the raw radioactivity values measured in the heart ROI in each individual animal were first averaged across animals within the group and then fitted using a linear interpolation before the curve peak and the sum of three decreasing exponentials after the peak. In each group, the brain ROI TACs measured in each individual animal were also averaged across animals within the group and fitted with a 2-tissue compartment model (2TCM) [26] using weights equal to the square root of each acquisition frame duration in order to provide estimates of the 2TCM rate constants (K_1_, k_2_, k_3_, and k_4_). The tracer total distribution volume (V_T_), the tracer non-displaceable distribution volume (V_ND_), and the tracer specific binding potential (BP_P_) in the brain ROI were then calculated from the rate constant estimates as V_T_ = K_1_/k_2_ × (1 + k_3_/k_4_), V_ND_ = K_1_/k_2_, and BP_P_ = K_1_/k_2_ × (k_3_/k_4_), respectively [27]. We also computed standard errors associated with the estimates of V_T_, V_ND_, and BP_P_. The analysis was also repeated using each individual heart curve and brain TAC instead of the group average curves. All analyses were performed using Matlab 2016b (www.mathworks.com).

### 2.9. Quantification and Statistical Analysis

Statistical analysis was performed using Prism 8.0 (San Diego, CA, USA). All data are represented as mean ± standard error. Statistical *p*-values were calculated using a two-tailed Student’s *t*-test for unpaired samples.

## 3. Results

### 3.1. Topotecan Demonstrates Variable Cytotoxicity between Human GBM Cells

Topotecan is a topoisomerase I inhibitor that demonstrates antitumor activity against a variety of human cancer cell lines [28] and potential activity in GBM after loco-regional delivery [29,30]. To evaluate the effective topotecan dose required in the brain, we exposed G48a and U251 GBM cells to various concentrations of topotecan (0.01 to 10 μg/mL). We found that topotecan is cytotoxic on both G48a and U251 cells but was noticeably more cytotoxic on U251 cells compared to G48a cells, with LD50s of 0.002 (0.001–0.003) μg/mL and 0.08 (0.06–1.0) μg/mL, respectively (Figure 1).

### 3.2. PET Imaging Confirms HIFU Permeabilizes the BBB

HIFU has the potential to significantly increase the concentration of topotecan in the brain and decrease systemic exposure. To confirm loco-regional BBB opening by HIFU and our ability to detect these effects in real-time using PET imaging, we treated mice with HIFU and imaged them with a surrogate PET tracer ([^18^F] FSPG), which has been shown not to penetrate an intact BBB both in animals [31] and in healthy volunteers [32]. To test the effect of HIFU on [^18^F] FSPG brain uptake, mice were treated with HIFU with microbubbles and then injected i.v. with [^18^F] FSPG. We found that [^18^F] FSPG did not penetrate the BBB and did not accumulate in the brains of untreated control mice (*n* = 4) in quantities detectable by PET (Figure 2a top). In contrast, after treatment with HIFU with microbubbles, the BBB was significantly permeabilized, as evidenced by [^18^F] FSPG PET (*n* = 4) (Figure 2a bottom). We further confirmed the HIFU-induced loco-regional opening of the BBB by obtaining PET images of dissected brains (Figure 2b). We confirmed HIFU-induced opening of the BBB by optical imaging of Cy7-albumin and Evan’s blue loco-regional accumulation in the brains of HIFU-treated mice, as we previously reported [33] (Figure 3). This strongly supports that HIFU with microbubbles opens the BBB, which can be detected using PET imaging.

### 3.3. HIFU Significantly Increases Loco-Regional Brain Concentration of Topotecan

To study the effects of HIFU on topotecan BBB permeability, we treated mice with HIFU with microbubbles and then immediately injected them i.v. with [^11^C] topotecan. PET images at 30 min post tracer injection demonstrated significantly higher levels of [^11^C] topotecan in the brains of HIFU-treated animals (*n* = 4) compared to control animals (*n* = 4) (Figure 4a), directly indicating real-time effects of HIFU on topotecan brain uptake. These data were confirmed in the postmortem PET images of dissected mouse brains (Figure 4b).

### 3.4. Kinetics of [^11^C] Topotecan in Mice Treated with HIFU

For kinetic analysis of topotecan, we obtained dynamic [^11^C] topotecan PET scans of control (*n* = 4) and HIFU-treated (*n* = 4) mice. We found lower levels of [^11^C] topotecan in the brain of control untreated mice compared to HIFU-treated mice, with a gradual decrease from a peak value of 0.88 ± 0.4% ID/g at 3 min post-injection to 0.2 ± 0.1% ID/g at the end of the scan (63 min post-injection) (Figure 5).

HIFU induced a three- to five-fold increase in the [^11^C] topotecan brain uptake, which peaked at 2.1 ± 0.7% ID/g at 3 min post-injection and gradually decreased to 1.1 ± 0.3% ID/g by the end of the scan. We used these data to analyze the kinetic rate parameters for [^11^C] topotecan injected into untreated mice (*n* = 4) and compared them to mice treated with HIFU (*n* = 4) (Figure 6). We found that [^11^C] topotecan kinetics in the brain are best described by a 2-tissue compartment model (2TCM) in comparison to a 1-tissue compartment and a 2-tissue-irreversible compartment models (Figure 6a). Kinetic data demonstrated a 2.2-fold higher V_T_ (volume of distribution) value of [^11^C] topotecan in the HIFU-treated mice (0.4 ± 0.02) compared to control mice (0.18 ± 0.02). Interestingly, both V_ND_ (the distribution volume of a non-displaceable [^11^C] topotecan) and BP_P_ (the distribution volume of specifically bound [^11^C] topotecan) were significantly increased after HIFU treatment, while K_1_ (rate constant for transfer from arterial plasma to the brain) was only moderately (1.3-fold) increased after HIFU (Figure 6). This suggests that HIFU significantly improves brain uptake of topotecan (both specific and non-specific) and increases the exposure time of topotecan in the brain, achieving considerably higher loco-regional topotecan concentration in the brain.

## 4. Discussion

HIFU with microbubbles is currently in clinical trials to permeabilize the BBB and allow efficient delivery of therapeutics into the brains of patients with intracranial malignancies [12]. This is especially beneficial for agents like topotecan that have significant systemic adverse effects and little or no BBB permeability. PET imaging can play a valuable role in quantifying BBB permeability as well as providing direct visualization of a radiolabeled drug. We confirmed the ability of PET to image HIFU-mediated BBB permeability using [^18^F] FSPG, as it does not cross the BBB in animals [31] or healthy volunteers [32] and has a relatively long half-life of 110 min. PET scanning after [^18^F] FSPG tracer administration was able to visualize and quantify BBB permeability in the area of the HIFU treatment (Figure 2). We confirmed these PET results by direct imaging of dissected brains and by EB dye localization. PET imaging, therefore, has potential use in the clinic to validate BBB opening and personalize therapeutic dose calculations for individual patients as HIFU technology continues to develop.

Topotecan is a topoisomerase inhibitor chemotherapeutic agent used for the treatment of small-cell lung carcinoma [16] and ovarian cancers [28], and sometimes for brain metastasis [18,19]. Earlier data suggested that topotecan could cross the BBB in normal rats [17], making it an attractive therapeutic agent to use for the treatment of brain malignancies. Our study demonstrated that topotecan is cytotoxic in G48a and U251 human glioblastoma cell lines (Figure 1). Moreover, PET imaging with [^11^C] topotecan tracer showed that it only crosses the BBB in healthy mice at a very low level (Figure 4, Figure 5 and Figure 6). [^11^C] topotecan brain concentration peaked at 0.9 ± 0.4% ID/g 3–5 min after injection followed by fast decline and plateaued at 0.2 ± 0.08% ID/g. This suggests that even though topotecan can minimally cross the BBB, one would need higher-than-optimal doses to achieve effective drug concentration where the BBB is intact.

Kinetic analysis is used to model tracer behavior and estimate tracer tissue uptake and clearance rates. We demonstrated that kinetic analysis can be used to estimate [^11^C] topotecan brain uptake and clearance in normal mice as well in mice treated with HIFU with microbubbles. In our study, we used HIFU with microbubbles to open the BBB to achieve higher topotecan concentrations in the brain. We demonstrated that injection of topotecan after HIFU at least doubles topotecan loco-regional peak concentration in the brain and significantly increases topotecan retention in the brain. Even after 1 h post-injection, we detected significant [^11^C] topotecan remaining in the brain (Figure 5), equal to topotecan peak concentration in the control mice. Based on our imaging data, 1–1.5% ID/gm of the topotecan dose was delivered to the mouse brain (Figure 4), primarily in the area of the HIFU application (Figure 4). The reported MTD of topotecan in GBM patients (1.5 mg/m^2^/d per day × five days [34]) implies that it would be feasible to administer 2.14 mg per day to a 70 kg person. Our imaging data suggest that the brain uptake, which predominantly takes place at the site of HIFU application, would receive a sustained dose of at least 20 μg/gm (1% of 2.14 mg). Our cell data demonstrate that cells in vitro had an LD_50_ significantly below this concentration (0.08–0.002 μg/mL). Furthermore, Kaiser et al., demonstrated that direct infusion of 2 µg/g/day for five days was effective at treating orthotopic GBMs [20]. Studying topotecan in combination with HIFU is justified because HIFU allows us to reach concentrations equal to or above these reported data. Our work strongly supports using HIFU to open the BBB before topotecan treatment in order to increase its therapeutic window and demonstrates the potential clinical application of [^11^C] topotecan PET as a tool to predict loco-regional brain concentration in patients with GBMs undergoing experimental HIFU treatments.

Promising preclinical data that indicate the safety and efficacy of multiple doses of ultrasound plus HIFU are emerging. For example, Wei et al., recently demonstrated that two doses of HIFU (7 days apart) plus microbubbles effectively made a subtherapeutic dose of etoposide efficacious against an orthotopic model of GBM [35]. Similar to topotecan, etoposide is a topoisomerase inhibitor that does not effectively cross the blood–brain barrier. While this is exciting, it indicates that for the application of HIFU with microbubbles to become a widespread means of opening the BBB for drug delivery, more work will likely have to be conducted looking at the effects of multiple applications over a short period of time, especially when treating GBM. Alternatively, it might be necessary to change the dosing regimen from daily in certain instances to less dense dosing (weekly or monthly). Importantly, as HIFU devices are becoming more advanced, they offer sufficient convenience and comfort to permit multiple applications. The most advanced image-guided ultrasound units set up in MRI scanners allow for the repeatability and comfort needed to perform these procedures on a routine basis.

## 5. Conclusions

We demonstrate that PET imaging is an effective tool to confirm BBB permeability and can potentially be used in patients with brain malignancies treated with HIFU. We also demonstrate the potential of modeling [^11^C] topotecan PET kinetics as a tool to predict regional topotecan brain concentration in the patients. This supports using HIFU plus microbubbles in conjunction with topotecan to potentially increase its therapeutic window in intracranial malignancies.

## Figures and Tables

**Figure 1 pharmaceutics-13-00405-f001:**
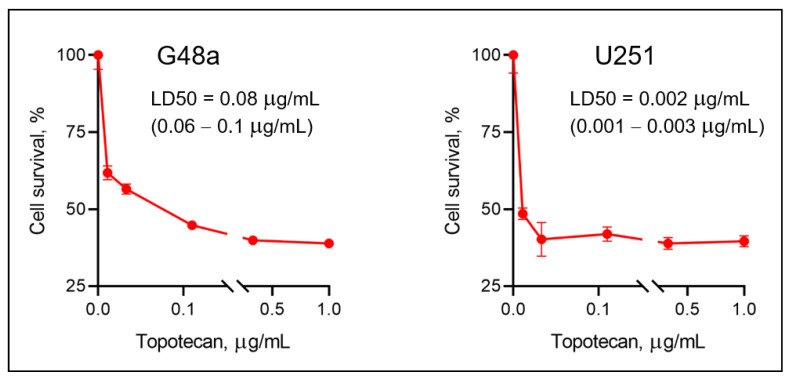
Cytotoxic effect of topotecan on G48a and U251 human glioblastoma cells. G48a and U251 cell cultures were treated with topotecan (0.012–1 μg/mL) and assayed for survival 24 h post-treatment (*n* = 4). Doses of topotecan inducing death of 50% of G48a and U251 cells (LD50) were calculated using GraphPad Prism software.

**Figure 2 pharmaceutics-13-00405-f002:**
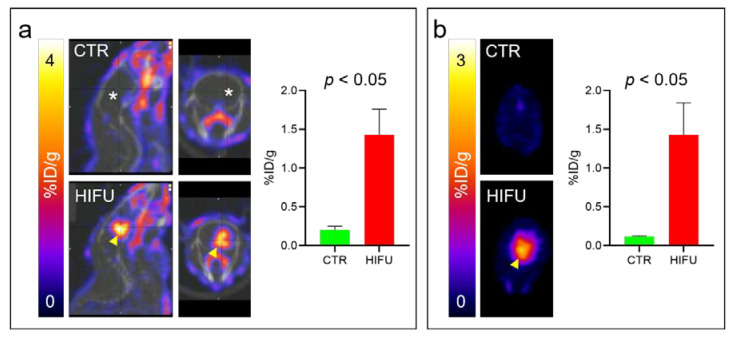
High-intensity focused ultrasound (HIFU) with microbubbles disrupts the blood–brain barrier (BBB) demonstrated with loco-regional brain uptake of [^18^F]-(4S)-4-(3-[18F]-fluoropropyl)-l-glutamic acid ([^18^F] FSPG) seen on positron emission tomography (PET) scan. (**a**) Control mice (*n* = 4) or mice treated with HIFU with microbubbles (*n* = 4) were injected i.v. with [^18^F] FSPG. Thirty minutes post tracer injection, static PET images were obtained. Representative PET images for control (CTR, no HIFU) and HIFU-treated (HIFU) mice are shown together with quantification of PET signal. (**b**) Immediately after PET imaging, the brains of all mice were dissected, imaged, and quantified with PET to confirm in vivo PET signal. Asterisk, no [^18^F] FSPG uptake detected in the brain of control mice. Arrowhead, [^18^F] FSPG signal in the brain of mice after HIFU.

**Figure 3 pharmaceutics-13-00405-f003:**
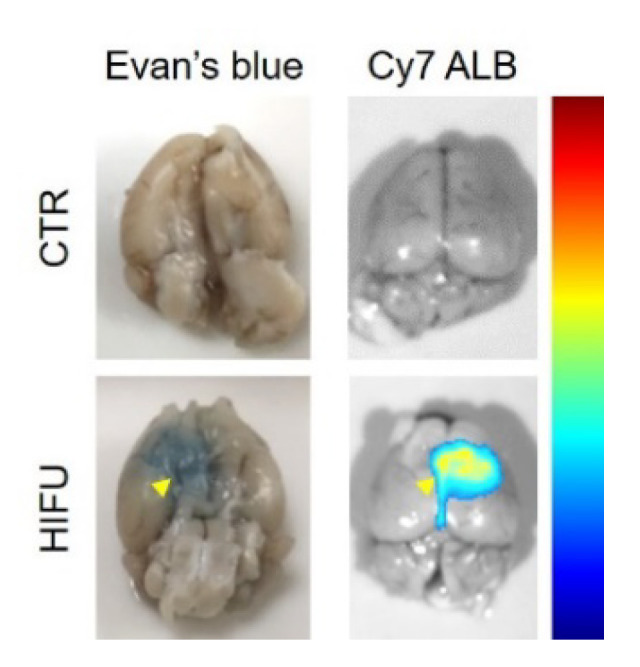
HIFU with microbubbles disrupts the BBB detected by accumulation of Evan’s blue (left panel) and Cy7-albumin (Cy7 ALB, right panel). Arrowhead, accumulation of Evan’s blue (EB) and Cy7-albumin (Cy7 ALB) in the brain of mice after HIFU.

**Figure 4 pharmaceutics-13-00405-f004:**
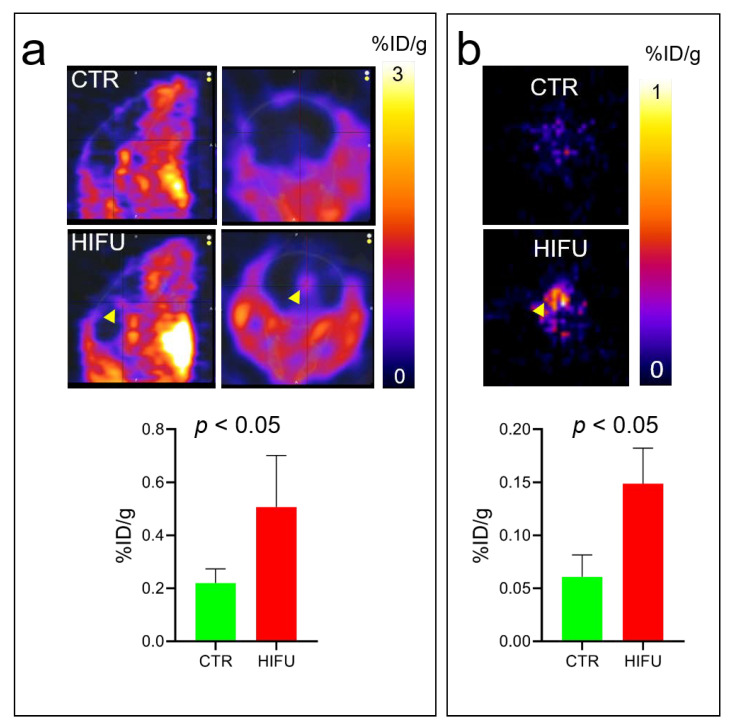
HIFU with microbubbles treatments permeabilize the BBB for topotecan and significantly increase the loco-regional accumulation of topotecan in the brain. (**a**) Representative PET images of [^11^C] topotecan uptake in brains of control (top panels) and HIFU-treated (bottom panels) mice. Images were quantified (*n* = 4) and demonstrated a significant increase in [^11^C] topotecan brain uptake after HIFU. (**b**) Immediately after PET imaging, the brains of all mice were dissected, imaged, and quantified with PET to confirm in vivo PET signal. PET images and quantification of [^11^C] topotecan uptake in dissected brains confirm PET results on live mice. Arrowhead, [^11^C] topotecan signal in the brain of mice after HIFU.

**Figure 5 pharmaceutics-13-00405-f005:**
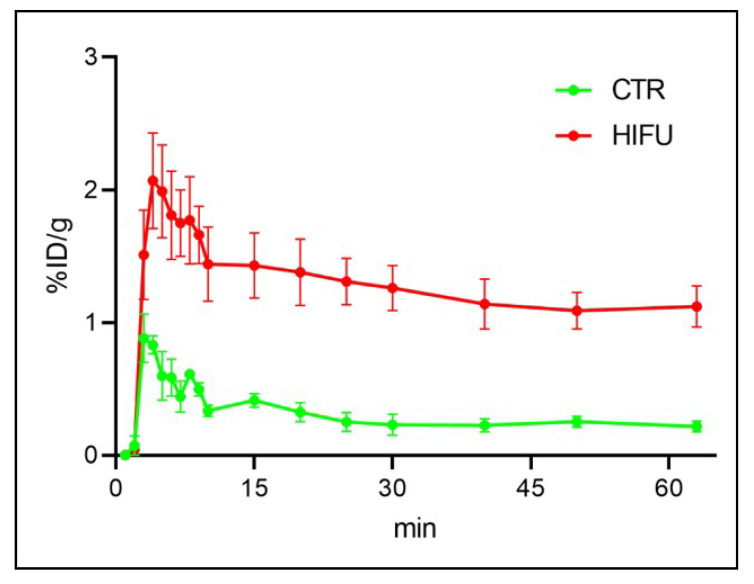
Dynamic PET scans of [^11^C] topotecan injected mice with (*n* = 4) and without HIFU treatment (*n* = 4) demonstrating increased peak and retained topotecan in the brain of HIFU-treated animals.

**Figure 6 pharmaceutics-13-00405-f006:**
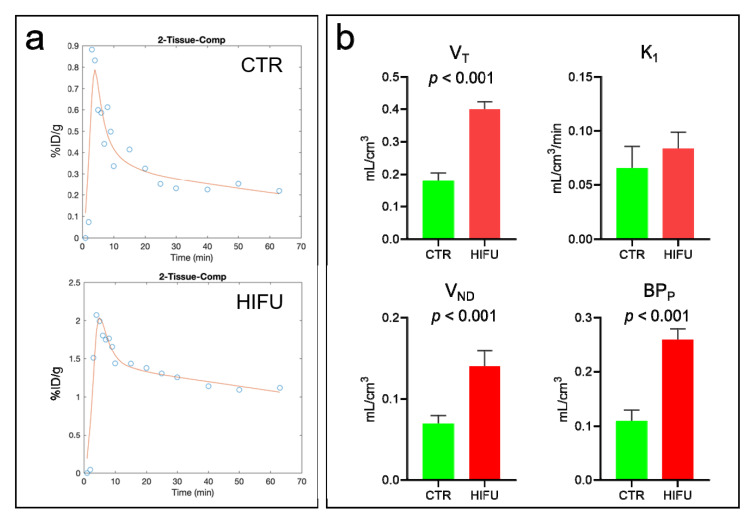
Kinetics of [^11^C] topotecan in mice after HIFU with microbubbles treatments. (**a**) 2TCM modeling of [^11^C] topotecan distribution in the control and HIFU-treated mice. (**b**) [^11^C] topotecan total distribution volume (V_T_), [^11^C] topotecan non-displaceable distribution volume (V_ND_), the distribution volume of specifically bound [^11^C] topotecan (BP_P_), and rate constant for [^11^C] topotecan transfer from arterial plasma to the brain (K_1_) in brains of control untreated and HIFU-treated mice. Note a significantly increased V_T_, V_ND,_ and BP_P_ of [^11^C] topotecan for the brains of mice treated with HIFU.

## Data Availability

Not applicable.

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
