# Peer review of "Real-Time Positron Emission Tomography Evaluation of Topotecan Brain Kinetics after Ultrasound-Mediated Blood–Brain Barrier Permeability"

_pharmaceutics, 2021, doi:10.3390/pharmaceutics13030405_

Round 1

Reviewer 1 Report

The authors have shown that HIFU can improve brain penetrance of topotecan, a well-documented cytotoxic agent for glioblastoma cells in vitro but with limited BBB permeability. The authors showed that the PET scans of isotope labeled Topotecan increased 3-to-5 folds in brain after HIFU. The time course of following the kinetics of PET imaging of topotecan nicely demonstrated the feasibility of continuing imaging of the compound, providing further value in preclinical testing of combination of treatment modality. Several issues that were raised in previous submission were addressed or adequately discussed. I support the publication of the manuscript.

Author Response

We thank reviewer 1 for the helpful comments and suggestions.

Reviewer 2 Report

The authors have considered previous reviewer comments and his reply was convincing to some extent. 

Author Response

We thank reviewer 2 for the helpful comments and suggestions.

Reviewer 3 Report

Molotkov et al. Describe in their manuscript an increase of topotecan mouse brain concentrations induced via HIFU with microbubbles which they quantified via PET imaging. While the experiments were well designed, and the results were essentially clearly presented.

A more critical discussion of the own data as well as the implication for topotecan treatment of intracranial malignancies is, however. needed:  

Compared to the substantial, ca. 7-11 fold, increase of FSPG brain exposure (Figure 2), the effect on topotecan was less than 3-fold (Figure 3) in the same experimental setting. As the manuscript is focusing on potential value for brain tumor therapy a scientific rational for these differnces should be given.

Topotecan has a moderate intrinsic permeability and is a BCRP substrates. Based on the described mechanism of opening the BBB one can expect an increase of k1, as demonstrated and also visible in Figure 4. One would also expect that k2 for a compound being a substrate of an efflux transporter is not decreased. Based on the model, k2 is, however reduced, which one would also not necessary expect from the plotted data in Figure 4. The concerns need be adressed.

The increase of k1 and decrease of k2 in your model with a parallel minor increase of k3 and k4 should result in an increase of C1, the free & non-specifically bound topotecan. One could argue that you see mainly an increase of non-specific binding which would not necessarily support the hope for a better efficacy. Can the second, more flat elimination phase seen in Figure 4 be interpreted into that direction?

The kinetic parameter of the 2TCM are point estimates. The extent of Vt change is in line with the less than 3-fold increase of exposure depicted in Figure 3 but not do the “up to 5-fold” change mentioned in the description of Figure 4. To judge the described, mostly quiet low, changes of the constants k1 to k4 it is important to understand the confidence into these values. Please provide those data and adopt the data interpretation where necessary.

The  line of argumentation for a potential therapeutic is not convincing and should be revised:

  • In vitro toxicity data, typically do not translate one to one into tumor shrinking in vivo. Clinically efficacious plasma concentrations for the treatment of peripheral tumors would be the appropriate starting point for this discussion. From my prospective section 3.1 and related text could be deleted.
  • The higher exposure – if considered sufficient for efficacy - was only demonstrated and discussed for the 1 hour observation period of the described experiments. Is that sufficient for efficacy? What are clinical data telling about duration of exposure needed to be efficacious in humans with respect to peripheral tumors?

Line 350-351 (conclusions): “This validates …”: The conclusion is not supported by the results of this study. Validation would need at least a proof of efficacy in a suitable in vivo model of intracranial malignancies. Sentence needs to be rephrased.

Minor

Figure 2 and 3: Please explain the meaning of the symbols (stars and arrows) depicted in the Figures

Confirmation of BBB opening by optical imaging of Cy7-albumin and Evan’s blue (Line 227-229):  Please add the results of the optical imaging with Cy7-albumin and Evan’s blue. In addition, the reference no. 33 is referring to a one page poster without data on Cy7-ablumin and Evan’s blue, please remove or correct.

Author Response

Dear Editor,

We would like to thank the reviewers for their helpful comments. Below are responses to reviewers’ feedback:

We thank reviewers 1 and 2 for the helpful comments and suggestions.

Reviewer 3

Compared to the substantial, ca. 7-11 fold, increase of FSPG brain exposure (Figure 2), the effect on topotecan was less than 3-fold (Figure 3) in the same experimental setting. As the manuscript is focusing on potential value for brain tumor therapy a scientific rational for these differences should be given.

We thank the reviewer for bringing this up. As the reviewer mentioned “topotecan has a moderate intrinsic permeability” of the intact BBB, while FSPG does not significantly penetrate the BBB in the amounts detectable by PET. The increased baseline topotecan level explains why we only see 3-fold increase on Figure 3.

Topotecan has a moderate intrinsic permeability and is a BCRP substrates. Based on the described mechanism of opening the BBB one can expect an increase of k1, as demonstrated and also visible in Figure 4. One would also expect that k2 for a compound being a substrate of an efflux transporter is not decreased. Based on the model, k2 is, however reduced, which one would also not necessary expect from the plotted data in Figure 4. The concerns need be addressed.

Thank you for the comment. Yes, topotecan is a substrate of an efflux transporter that will affect topotecan kinetics through the intact non-damaged BBB. However, we use HIFU with microbubbles to damage BBB and significantly increasing its permeability. One can assume that this will significantly decrease BCRP contribution to the topotecan kinetic in the brains of mice after HIFU, which is part of the new information provided by using topotecan PET imaging.

The increase of k1 and decrease of k2 in your model with a parallel minor increase of k3 and k4 should result in an increase of C1, the free & non-specifically bound topotecan. One could argue that you see mainly an increase of non-specific binding which would not necessarily support the hope for a better efficacy. Can the second, more flat elimination phase seen in Figure 4 be interpreted into that direction?

We thank the reviewer for the comment. The higher tracer total distribution volume (VT) in presence of HIFU could be due to higher concentration of free/non-specifically bound topotecan, higher concentration of specifically bound topotecan, or a combination of both. To answer this question, we calculated the tracer specific distribution volume (also known as the binding potential BPp) as BPp = K1/k2*(k3/k4) [1], to estimate specific binding of topotecan in the brain of control mice and mice after HIFU (mean + standard error). We found that BPp was ~2.4-folds higher in the brains of mice after HIFU (0.26+0.03) when compared with untreated mice (0.11+0.02). This suggests that the increase of topotecan brain uptake after HIFU treatment that we demonstrated using PET scan leads to increase of the topotecan specific binding. We included these new data in the manuscript.  

The kinetic parameter of the 2TCM are point estimates. The extent of Vt change is in line with the less than 3-fold increase of exposure depicted in Figure 3 but not do the “up to 5-fold” change mentioned in the description of Figure 4. To judge the described, mostly quiet low, changes of the constants k1 to k4 it is important to understand the confidence into these values. Please provide those data and adopt the data interpretation where necessary.

We thank reviewer for the comment. We calculated the standard error for all the kinetic parameters, as described in Ogden RT, Tarpey T (2006) [2], and included these data in the manuscript. We also changed interpretation where needed.

The line of argumentation for a potential therapeutic is not convincing and should be revised:

In vitro toxicity data, typically do not translate one to one into tumor shrinking in vivo. Clinically efficacious plasma concentrations for the treatment of peripheral tumors would be the appropriate starting point for this discussion. From my prospective section 3.1 and related text could be deleted.

We agree that in vitro toxicity data may not be directly translated into clinical data, but felt that if topotecan did not kill GBM cells in vitro, it would definitely not work in vivo. We therefore performed initial confirmatory in vitro cytotoxicity. Therefore, we believe that this data contribute meaningfully to the manuscript and should be included.   Furthermore, topotecan has been used in locoregional delivery preclinical and clinical trials, demonstrating potential future use in the CNS if it can be delivered in sufficient quantity.

The higher exposure – if considered sufficient for efficacy - was only demonstrated and discussed for the 1 hour observation period of the described experiments. Is that sufficient for efficacy? What are clinical data telling about duration of exposure needed to be efficacious in humans with respect to peripheral tumors?

We thank reviewer for the comment. Clinical data shows that the standard-of-care course of treatment for topotecan chemotherapy is  5 consecutive injections (single injection per day) that can be repeated for several months. Due to technical reasons these treatment regimens are not possible in experiments using mice as a model. Nevertheless, our data clearly demonstrate that HIFU with microbubbles significantly increases the level of the topotecan in the brain. This provides direct preliminary evidence of increased topotecan levels in the brain, which justifies further studies to evaluate if topotecan plus HIFU can potentially achieve either better efficacy and/or lower toxicity than systemic topotecan.

Line 350-351 (conclusions): “This validates …”: The conclusion is not supported by the results of this study. Validation would need at least a proof of efficacy in a suitable in vivo model of intracranial malignancies. Sentence needs to be rephrased.   

Currently, topotecan is used in clinic for the treatment of the brain metastasis. Our study further demonstrates that combination of the topotecan with HIFU allows for significantly increased level of the topotecan intracranial concentration. We changed the language to reflect the reviewers point of using the work validate other than in the context of increasing topotecan brain delivery after HIFU administration. 

Minor points:

Figure 2 and 3: Please explain the meaning of the symbols (stars and arrows) depicted in the Figures

We thank reviewer for catching this. We added description to the Figure legends.

Confirmation of BBB opening by optical imaging of Cy7-albumin and Evan’s blue (Line 227-229):  Please add the results of the optical imaging with Cy7-albumin and Evan’s blue.

We thank reviewer for that comment. We included data in the manuscript as Figure 3.

In addition, the reference no. 33 is referring to a one page poster without data on Cy7-ablumin and Evan’s blue, please remove or correct.

We thank reviewer for catching this. We corrected the reference.

Citations

  1. Innis, R.B., et al., Consensus nomenclature for in vivo imaging of reversibly binding radioligands. J Cereb Blood Flow Metab, 2007. 27(9): p. 1533-9.
  2. Ogden, R.T. and T. Tarpey, Estimation in regression models with externally estimated parameters. Biostatistics, 2006. 7(1): p. 115-29.

Round 2

Reviewer 3 Report

I appreciate the authors modifications in the results section and the rephrasing of the conclusion. 

This manuscript is a resubmission of an earlier submission. The following is a list of the peer review reports and author responses from that submission.

Round 1

Reviewer 1 Report

The authors have shown that HIFU can improve brain penetrance of topotecan, a well-documented cytotoxic agent for glioblastoma cells in vitro but with limited BBB permeability. The authors showed that the PET scans of isotope labeled Topotecan increased 3-to-5 folds in brain after HIFU. The time course of following the kinetics of PET imaging of topotecan nicely demonstrated the feasibility of continuing imaging of the compound, providing further value in preclinical testing of combination of treatment modality.

The one important aspect that is currently lacking is how to assess if the brain penetrance of topotecan after HIFU is of importance. The authors have not shown any results in combining topotecan with HIFU in intracranially implanted PDX models. There is also no discussion if the increased concentration in brain uptake is sufficient in reaching cellular IC50 that were found in figure 1. Therefore it is not yet clear to the readers if such combination modality will be of value. The limitation of HIFU in clinical development is often the frequency of the procedures. This should be discussed if such combination of HIFU with topotecan is used and what would be the next steps in addressing these questions.

Reviewer 2 Report

The present review aims to provide study Real-time positron emission tomography evaluation of topotecan brain kinetics after ultrasound-mediated blood brain barrier permeability.

The is a very hot topic especially its potential application to overcome the BBB, and to allow a more effective penetration/diffusion mechanism by which targeted therapy for brain malignancies. However, in order to have a clinical relevance of the envisaged modality for brain malignancy, the inclusion of in vivo studies to evaluate the therapeutic potential of topotecan after

 HIFU manipulation would increase the clinical relevance and the importance of this modality. Moreover, the inclusion of some genomic/proteomic markers to highlight the underlying molecular mechanism by which HIFU can alter the permeability of BBB, and by which combined treatment of HIFU + topotecan can modulate the growth of brain cancer would be timely and of great importance to the field of brain malignancies. Therefore, we encourage the authors to add the experiments with the aim to strengthen the clinical relevance of the presented observations/outcomes.